# Keeping the "R" in LARC (long-acting reversible contraception): Measuring client-centered implant removal services in sub-Saharan Africa

Celia Karp[1]*, Katherine Tumlinson[2], Brooke W. Bullington[3], Linnea A. Zimmerman[1], Leigh Senderowicz[4]

**1** Department of Population Family and Reproductive Health, Johns Hopkins Bloomberg School of Public Health, Baltimore, Maryland, United States of America, **2** Carolina Population Center and Department of Health Planning and Management, Gillings School of Global Public Health, University of North Carolina at Chapel Hill, Chapel Hill, North Carolina, United States of America, **3** Department of Epidemiology and Carolina Population Center, Gillings School of Global Public Health, University of North Carolina at Chapel Hill, Chapel Hill, North Carolina, United States of America, **4** Department of Gender and Women's Studies, University of Wisconsin-Madison, Madison, Wisconsin, United States of America

* celia.karp@jhu.edu

## Abstract

As the use of subdermal contraceptive implants increases across sub-Saharan Africa, the need for person-centered removal services is more critical than ever to safeguard reproductive autonomy. In 2016, Christofield and Lacoste proposed eight conditions for client-centered implant removal, yet the extent to which these conditions have been assessed in large-scale surveys remains unexamined. Our mapping exercise collates survey information from three large data collection platforms fielded in sub-Saharan Africa, including the Demographic and Health Surveys (DHS), Performance Monitoring for Action (PMA), and the Service Availability and Readiness Assessment (SARA), utilizing questionnaire tools implemented among women, health facilities, providers, and clients to map existing data sources against these conditions. Our findings reveal that four of the eight conditions are fully captured, three are partially captured, and one is entirely absent within current population or facility-based surveys. Specifically, among the six facility-based conditions: the three conditions completely captured include the availability of supplies (condition 2), systems for managing difficult removals (condition 3), and the provision of counseling and reinsertion options (condition 5); two conditions partially captured include competent and confident providers (condition 1) and affordability for clients (condition 7), and the one condition that remained absent was measurement systems for monitoring of removal data (condition 8). Among the two individual-level conditions, timely and proximate service (condition 4) was partially captured and knowledge and awareness of where to go for removal (condition 6) was fully captured. Nearly a decade after Christofield and Lacoste proposed these client-centered conditions, the lack of consistent measures for client-centered implant removal poses significant barriers to

**Data availability statement:** Data used in this study are derived from publicly available survey instruments accessible from the Demographic and Health Surveys (DHS) via https://www.dhsprogram.com/, Performance Monitoring for Action (PMA) via https://www.pmadata.org/, and the Service Availability and Readiness Assessment (SARA) via https://www.who.int/data/data-collection-tools/service-availability-and-readiness-assessment-(sara).

**Funding:** This work was supported, in whole or part, by the Bill & Melinda Gates Foundation IVN 009639. Under the grant conditions of the Foundation, a Creative Commons Attribution 4.0 Generic License has already been assigned to the Author Accepted Manuscript version that might arise from this submission. Contributions by CK and LZ, were supported by grants OPP1198333 and OPP1198339 awarded by the Bill & Melinda Gates Foundation. Support for this research was also provided, in part, by a career development grant (R00 HD086270) to KMT and an infrastructure grant for population research (P2C HD050924) to the Carolina Population Center at the University of North Carolina at Chapel Hill, both from The Eunice Kennedy Shriver National Institute of Child Health and Human Development (NICHD) of the National Institutes of Health (NIH). LS's contribution was supported by a Ruth L Kirschstein National Research Service Award (T32 HD049302) and Population Research Infrastructure grant (P2C HD047873) from the NICHD. BB's contribution was supported in part by funding from NICHD (T32 HD052468) and Population Research Infrastructure grant (P2C HD050924). The funders had no role in the study design, data collection and analysis, decision to publish, or preparation of the manuscript.

**Competing interests:** The authors have declared that no competing interests exist.

understanding service accessibility and women's reproductive experiences. To protect the reproductive autonomy of individuals who use long-acting reversible contraceptive (LARC) methods and desire timely discontinuation, it is imperative to develop and implement standardized metrics for their removal.

## Background

Over the last two decades, use of subdermal contraceptive implants has grown dramatically, particularly in sub-Saharan Africa. In countries as diverse as Burkina Faso, Ethiopia, Democratic Republic of the Congo, Ghana, Kenya, and Senegal, implants currently account for one-fourth to one-half of all modern contraceptive use [1,2]. Evidence from 2021 indicates that implants have become the most or second-most common method in 20 African countries [3]. Contraceptive users often cite implants as a preferred method due to their extended duration of protection (3–5 years), potential for covert or concealed use from partners, family members, and other actors who may influence contraceptive decisions, and fewer perceived side effects relative to other methods, such as injectables and pills [4,5].

As the number of implant users continues to rise, however, so too does the need for implant removal services. Modeled projections of implant removal needs based on method procurement data from the Reproductive Health Supplies Coalition suggested the number of implant removals needed in 2018 were estimated to be more than double that of 2015 [6]. Trends in implant use underscore this lagged increase in demand for implant removal services, requiring equipped facilities and trained, willing providers to ensure individuals can discontinue their contraceptive implant when desired. At the same time, a growing body of qualitative evidence suggests many women face considerable barriers to accessing implant removal services, including cost of user fees for removal, lack of geographic access to providers or facilities offering removal, provider reluctance to remove implants "early" (i.e., before the device's expiration date), and provider hesitance due to lack of training, experience, or appropriate removal equipment [7–15]. These barriers to removal threaten the reproductive autonomy of implant users, inhibiting their ability to switch methods or stop using contraception altogether. Despite this documentation of obstacles for implant users who desire removal, the sexual and reproductive health field lacks systematic, quantitative measurement of access to client-centered implant removal at both the individual and health systems levels, limiting opportunities for monitoring and improvement of services.

In 2016, Christofield and Lacoste proposed a list of eight conditions necessary for client-centered implant removal, including recommendations for ensuring future indicators align with principles of high-quality, person-centered care [6]. Two conditions relate to women's knowledge about removal services (e.g., knowing where and when to seek removal), and six conditions reflect aspects of facility and provider environments (e.g., supplies and equipment in-place, trained providers). While the conditions proposed by Christofield and Lacoste offer a useful framework for evaluating and

strengthening implant removal services, to date, examination of these conditions using existing, large-scale data collection platforms for reproductive health has not been assessed. Operationalizing measurement of health system readiness to provide quality-centered implant removal, access to care, and knowledge of removal services among potential clients will improve knowledge and inform interventions that enhance access to and delivery of high-quality implant removal services.

In this analysis, we apply the conditions of the Christofield and Lacoste framework to examine measurement of client-centered implant removal in existing data collection efforts, including population and facility-based health surveys. Specifically, we aimed to explore opportunities and gaps in measurement of conditions for quality-centered implant removal services in sub-Saharan Africa by mapping existing data sources for the region onto the Christofield and Lacoste framework.

## Methods

### Study design

We implemented an observational, text-based mapping exercise of survey items-to-indicators across survey instruments to examine coverage of client-centered implant removal conditions in existing surveys of health facilities and women implemented in sub-Saharan Africa.

### Survey instruments

We used survey instruments from three large-scale data collection platforms that implement surveys among women, health facilities, health workers, and/or family planning clients in low- and middle-income countries (LMICs). Details of each data source are provided in Table 1. In brief, survey instruments included the 1) female survey, service delivery point (SDP) survey, and client exit interview (CEI) of the Performance Monitoring for Action (PMA) project; 2) the core facility instrument of the Service Availability and Readiness Assessment (SARA), implemented by the World Health Organization (WHO); and 3) the facility inventory, health worker interview, observation of family planning consultation tool, and CEI of the Service Provision Assessment (SPA) implemented by Demographic and Health Surveys (DHS), as well as the DHS program's women's survey. Altogether, nine unique survey instruments were examined. SPA and SARA surveys generate nationally representative data of availability, readiness, and quality of health systems in a wide range of LMICs and are conducted approximately every five years. The DHS program has implemented surveys in over 90 countries since its inception in 1984, adding the facility-based SPA survey in 1999; the SPA has since been fielded in 18 countries. PMA's two facility-based surveys (SDP and CEI) are implemented annually in 10 countries across sub-Saharan Africa and south Asia alongside a nationally representative household-based survey of women aged 15–49 years (female survey). While PMA's SDP survey does not generate nationally representative estimates, data reflect the health service environment of households sampled. Together, the three large-scale data collection projects—DHS, PMA, and SARA—offer key insights into a range of population, health, and service availability indicators across LMICs, including many in sub-Saharan Africa.

For our mapping exercise, we used the most recent, publicly available survey instruments accessible online for each data collection platform as of 2024. In 2022, the SPA questionnaires were redesigned to explicitly orient measurement toward person-centered indicators. The new SPA surveys were not yet implemented at the time of this analysis; however, the new questionnaires included in the Core Phase 8 (DHS8), which were publicly available online and included in this analysis [16]. Similarly, we used the most recent female, SDP, and CEI questionnaires available from PMA's Phase 3 surveys, implemented between 2021–2023 [17]. Finally, we relied on the SARA questionnaires from 2014, which were available on the WHO website [18].

### Mapping of client-centered conditions

We began by mapping individual survey questions from the included PMA, SARA, and DHS/SPA questionnaires to each of the conditions for client-centered implant removal, as outlined by Christofield and Lacoste (Table 2). The

**Table 1. Survey instrument and sampling description by data source.**

| Organization | Data Source | Survey Instrument(s) | Description of Population and Sampling |
|---|---|---|---|
| William H. Gates Institute Sr. Institute for Population and Reproductive Health | Performance Monitoring for Action (PMA) | Female survey | Nationally or sub-nationally representative sample of women aged 15-49 from selected communities identified and interviewed using a household-based survey and roster |
| | | Service delivery point survey | Probability sample of health facilities, pharmacies, and retail outlets that offer family planning (FP) services to the selected communities |
| | | Client exit interview survey | Systematic selection of approximately 10 clients per service delivery point facility. Eligible clients are females aged 18-49 years who received FP services on the day of interview |
| World Health Organization (WHO) | Service Availability and Readiness Assessment (SARA) | Facility survey | Nationally or sub-nationally representative sample obtained from a simple random sample of facilities within each stratum (facility type and managing authority) at the national or sub-national level |
| Demographic and Health Surveys (DHS) Program | Service Provision Assessment (SPA) | Facility inventory | National sample of formal sector health facilities selected from a comprehensive list of health facilities in a country (sampling frame), categorized by facility type, managing authority (public and non-public), and by region. Pharmacies and individual doctors' offices are typically not included in a SPA. |
| | | Health worker interview | Convenient random sampling of all providers linked to a selected health facility who are present on the day of the survey and provide antenatal care (ANC), FP, or sick child care. |
| | | Client exit interview | Convenient sample of all clients waiting to see selected provider who offers ANC, FP, or sick child services. For FP, new FP clients are prioritized and up to eight ("non-priority", i.e., not new FP) clients are observed for each provider. All observed clients or caretakers of sick children are eligible for exit interviews before they leave the facility |
| | | Observation of Family Planning Consultation | Interviewers observe client-provider consultations for three priority services using Observation Protocols: antenatal care (ANC), family planning (FP), and curative care for sick children. This is to assess how often consultations follow generally accepted standards of care, including integrated management of child illnesses (IMCI). The number of consultations observed depends on the number of providers and clients in the facility that day. |
| | Demographic and Health Surveys (DHS) Program | Woman's survey | Nationally and sub-nationally representative sample of women aged 15-49 from selected communities identified and interviewed using a household-based survey and roster |

two individual-level conditions included service available when she wants, within reasonable distance (condition 4); and woman knows where and when to go for removal (condition 6); availability of survey data for these conditions was assessed using PMA's female survey and DHS's woman's survey. The six facility-based conditions included availability of competent and confident providers (condition 1); supplies and equipment in-place (condition 2); system in-place for managing difficult removals (condition 3); reassurance, counseling, and reinsertion/switching offered (condition 5); service is affordable or free (condition 7); and implant removal data collected and monitored (condition 8). Some survey items from non-facility-based surveys (e.g., women's survey of DHS, female questionnaire of PMA) were considered as potential measures for facility-based conditions, where appropriate. For each condition, we proposed an operational definition for measuring the condition, based on conceptualization developed by Christofield and Lacoste and later expanded by the Implant Removal Task Force. Next, we assessed alignment between each condition and the data collected by PMA, SARA, and DHS/SPA surveys via mapping of individual survey items. Finally, we assessed the extent to which each client-centered implant removal condition was captured within each survey tool by assigning a status of completely captured, partially captured, and not captured to each condition, by survey instrument.

**Table 2. Conditions to assess client-centered implant removal, operational definition, and level for data collection.**

| Condition | Operationalization | Data source |
|---|---|---|
| 1. Competent and confident provider | Facility has providers available with capacity to provide implant removal. Provider competence may be measured at the provider or facility levels by recent training completion, direct observation of care, or survey questions about the procedure. Provider confidence may be measured using a health worker survey measuring self-efficacy in implant removal procedures. | Provider interview or Facility survey |
| 2. Supplies and equipment in place | Facility is equipped with necessary supplies and equipment for implant removal procedures. Where applicable, the facility has supplies and equipment available for deeply placed or non-palpable removals.<br>Essential supplies and equipment for implant removal adapted from Engender Health "Long-Acting Reversible and Permanent Methods of Contraception Supplies List" include sterile gloves, antiseptic, sterile gauze sponges or cotton wool, local anesthetic (e.g., lidocaine), scalpel/surgical blade, forceps (straight/curved) | Facility survey |
| 3. System in place for managing difficult removals | Facility has a system for managing non-palpable and difficult implant removals. Systems may include referral services to higher-level facilities. | Facility survey |
| 4. Service available when she wants, within reasonable distance (*timely and proximate service available*) | Implant removal services are regularly available and accessible to reduce potential barriers to care occurring within service delivery points, including static and mobile service delivery channels. Service availability may be measured at the individual or facility level. At the individual level, service availability may be measured via client exit interviews that ask clients about problems accessing care, such as time, cost, and hours of operation at facilities or in population-based surveys that ask about perceived access to points of care for implant removal. At the facility level, service availability may be measured via health facility surveys that record days, times, and regularity of implant removal services, as these may only be offered on specific days when trained providers are available. | Individual client interview and Facility survey |
| 5. Reassurance, counseling, and reinsertion/switching are offered | Providers equip potential implant users with information about side effects, follow-up care, and method management. At the individual level, information from individual surveys via client exit interviews that ask clients about their experiences of contraceptive counseling. Similar information about counseling experiences can be obtained from direct observations of family planning consultations. | Individual client interview, Observation of counseling interaction, or Facility survey |
| 6. Client knows where and when to go for removal | Implant users are knowledgeable about where to access implant removal services and when to seek care for removal. At the individual level, information about timing and location for implant removal services may be measured through client exit interviews or population-based surveys to assess where and how implant users receive, understand, and act on this information. | Individual client interview |
| 7. Service is affordable or free | Implant removal services are offered for free or at a cost that is acceptable to implant users. Removal service costs may be measured at the individual or facility levels. At the individual level, service affordability may be measured by asking clients about costs they paid for care and whether the cost was a problem. At the facility level, service affordability may be measured by asking about costs related to family planning services and implant removal specifically. | Individual client interview or Facility survey |
| 8. Implant removal data collected and monitored | Data about implant removal services is collected regularly as part of routine monitoring systems. Implant removal data may be measured at the individual or facility level. At the individual level, removal data can be collected by measuring women's reports of recent implant use and asking about timing and experience of removal. At the facility level, removal data may be measured through direct reporting in HMIS/DHIS2 or facility registers, which can be integrated into facility-based surveys for calculating implant removal caseloads. | Facility survey |

First, availability of competent and confident providers (condition 1) was defined as "providers available with capacity to provide implant removal". Provider competence was evaluated for potential measurement at the provider or facility levels by recent training completion, direct observation of care, or survey questions to providers about the procedure itself. Next, we assessed facility supplies and equipment in-place for removal (condition 2) by applying an adapted list of essential clinic supplies generated by Engender Health and recommended by the Implant Removal Taskforce [19]. Six essential supplies were included: clean gloves, antiseptic, sterile gauze pad or cotton wool, local anesthetic, surgical blade, and mosquito forceps. Availability of individual supplies and the complete supply list were assessed by survey instrument to determine the extent to which this condition was measured. System readiness for management of clients with deeply

placed implants (condition 3) was assessed according to the facility's capacity to remove and refer women for deeply placed, non-palpable implants. The delivery of reassurance, counseling, and reinsertion/switching related to contraceptive implants (condition 5) was defined as "providers equip implant users with information about side effects, follow-up care, and method management, such as the option to switch methods". Information about this provider-client interaction could be ascertained via client exit interviews that ask individuals about their experiences of contraceptive counseling or through direct observations of family planning consultations. Service affordability (condition 7) was defined as removal services being offered for free or at a cost that was "acceptable" to implant users. Removal service costs could be measured at the individual or facility levels. Finally, implant removal data collection and monitoring (condition 8) was defined as the regular or systematic collection of implant removal services data within a facility, which could be measured at the facility level.

This observational indicator mapping exercise included analysis of extant survey instruments for several large-scale surveys. The analysis did not include human subjects, nor the secondary analysis of data collected amongst human subjects, and, therefore, did not require approval from an institutional review board or ethics committee.

## Results

Of the eight conditions of quality-centered implant removal services proposed by Christofield and Lacoste, four were completely captured, three were partially captured, and one was not captured in any of the nine population-based or facility-based survey instruments examined. Coverage of each quality-centered implant removal condition is synthesized in Table 3 and detailed in Table 4.

### Individual-level conditions

Survey instruments included limited information about the two individual-level conditions for quality-centered implant removal, resulting in one condition that was partially captured and one that was fully captured by any of the nine survey instruments (Table 4). Condition 4—timely and proximate removal service available—was partially captured by survey questions asked in the client exit interviews (CEIs) administered by the SPA and PMA surveys. This condition was categorized as partially captured given the lack of direct questions about what a client considered to be a "reasonable" distance and whether the timing of the service aligned with the client's preferences for care. Instead, data collected through survey items that partially captured condition 4 include: among the sample of women who sought implant removal (i) if the facility where care was received was the closest one to the client's residence, (ii) transport duration and modality for arriving at the facility, (iii) reason for not seeking care at facility closest to one's residence, and (iv) client-perceived problem of facility

Table 3. Synthesis of metrics to assess quality-centered implant removal.

| Condition | DHS/SPA | PMA | SARA |
|---|---|---|---|
| 1. Competent and confident provider available | ✓- | ✓- | ✓- |
| 2. Supplies and equipment in-place | ✓ | ✓ | ✓ |
| 3. Systems in-place for difficult removals | x | ✓ | x |
| 4. Timely and proximate service available | ✓- | ✓- | x |
| 5. Reassurance, counseling, and reinsertion or switching | ✓ | ✓ | x |
| 6. Client knows when and where to go for removal | x | ✓ | x |
| 7. Service affordable or free | ✓- | ✓- | x |
| 8. Implant removal data collected and monitored | x | x | x |

*Notes: DHS = Demographic and Health Surveys, SPA = Service Provision Assessment, PMA = Performance Monitoring for Action, SARA = Service Availability and Readiness Assessment.

**Key** ✓ Complete ✓- Partial x Not captured

**Table 4. Survey items available in extant data sources to assess quality-centered implant removal conditions, by survey questionnaire.**

| Condition | DHS/ SPA | PMA | SARA |
|---|---|---|---|
| 1. Competent and confident provider | *Health Worker Interview* | *Service Delivery Point Survey* | Q704. Have you or any provider(s) of family planning services: |
| | Q401. Have you received any in-service training, training updates, or refresher training on topics related to family planning? | Q411. If a woman came today needing her implant removed, could that service be provided to her today onsite? | - 01. Received any family planning training in the last two years? |
| | Q402. Have you received any in-service training, training updates, or refresher training in any of the following topics [READ TOPIC] | Q412. If a woman comes to your facility today needing her implant removed, but it is deeply placed, could that service be provided to her today onsite? | |
| | IF YES: Was the training, training update, or refresher training within the past 24 months or more than 24 months ago? | | |
| | Q402.03. Implant insertion and/or removal? | | |
| 2. Supplies and equipment in place | *Facility Inventory* | *Service Delivery Point Survey* | Q600. Please tell me if the following resources/supplies used for infection control are available in the general outpatient area of this facility today. |
| | Q710.Standard precautions and conditions for client examination | Q411. If a woman came today needing her implant removed, could that service be provided to her today onsite? | - 03. Alcohol based hand rub<br>- 04. Disposable latex gloves<br>- 05. Waste receptacle<br>- 06. Sharps container ("safety box") |
| | - 02. Hand-washing soap<br>- 03. Alcohol-based hand rub<br>- 04. Waste receptacle with lid<br>- 06. Sharps container ("safety box")<br>- 07. Disposable latex gloves | Q412. If a woman comes to your facility today needing her implant removed, but it is deeply placed, could that service be provided to her today onsite? | Q2102. Please tell me if the following surgical equipment and supplies are available and functional in this facility today. |
| | Q1621. Do you have the following items? If yes, I would like to see them. (*Collected in facilities offering labor and delivery care) | Does this facility have the following supplies needed to insert and/or remove implants: | - 04. Scalpel handle with blades<br>- 06. Surgical scissors |
| | - 04. Scissors or blade to cut cord<br>- 08. Forceps (Large)<br>- 09. Forceps (Medium) | Read out all supplies and select all that apply. Supplies do not need to be observed, but must be available on the day of the interview. | Q2104. Please tell me if any of the following materials or medicines are available in this service site today. I would like to see those that are available. |
| | | - Clean Gloves<br>- Antiseptic<br>- Sterile Gauze Pad or Cotton Wool<br>- Local anesthetic<br>- Sealed Implant Pack<br>- Surgical Blade<br>- Mosquito forceps (straight or curved) | - 02. Skin disinfectant<br>- 04. Lidocaine 1% or 2% (anaesthesia) |
| 3. System in place for managing difficult removals | Not captured | *Service Delivery Point Survey*<br>Q413. Would someone at this facility know where to send her to have the implant removed? | Not captured |
| 4. Service available when she wants, within reasonable distance | *Client Exit Interview* | *Client Exit Interview* | Not captured |

*(Continued)*

**Table 4.** (Continued)

| Condition | DHS/ SPA | PMA | SARA |
|---|---|---|---|
| | Q301. Was the time you waited to see a provider a problem?<br><br>Q302. Were the hours of service at this facility, that is when the facility opens and closes, a problem?<br><br>Q303. Were the number of days services are available to you at this facility a problem?<br><br>IF YES, PROBE: Would you say this was a major problem or a minor problem?<br><br>Q305. Is this the closest health facility to your home?<br><br>Q306. What was the main reason you did not go to the facility nearest to your home?<br><br>IF CLIENT MENTIONS SEVERAL REASONS, PROBE FOR THE MOST IMPORTANT, OR MAIN REASON. | Q107. Is this the closest health facility to your current residence?<br><br>Q108. What was the main reason you did not go to the facility nearest to your home?<br><br>Q109. How much time did it take you to travel here today?<br><br>Q110. What means of transportation did you use to travel here? | |
| 5. Reassurance, counseling, and reinsertion/switching are offered | *Client Exit Interview*<br>Q201.05. Talk about possible side effects or problems with the method you selected?<br>Q201.06. Tell you what to do if you experience any side effects or problems with the method you selected?<br><br>Q201.07. Talk about warning signs associated with the method you selected?<br><br>Q201.08. Talk about the possibility of switching to another method if the method you selected was not suitable?<br>*Woman's Questionnaire*<br><br>Q323. At that time, were you told about side effects or problems you might have with the method?<br><br>Q325. Were you told what to do if you experienced side effects or problems?<br>Q326. At that time, were you told about other methods of family planning that you could use?<br>Q328. At that time, were you told that you could switch to another method if you wanted to or needed to? | *Client Exit Interview*<br>Q217. I felt encouraged to ask questions and express my concerns.<br>Q219. The provider asked me questions in order to provide counseling that fit me personally.<br>Q220. I received all of the information I wanted to know about my options for contraceptive methods.<br>Q222. After this consultation, I could understand how my body might react to using contraception.<br>Q223. I could understand how to use the method(s) we talked about during the consultation.<br>Q228. During your visit today, were you told by the provider about advantages and disadvantages with a method to delay or avoid pregnancy?<br>Q229. What advantages did the provider tell you about your [METHOD]?<br>Q230. What disadvantages did the provider tell you about your [METHOD]? | Not captured |
| | *Observation of Family Planning Consultation*<br>Q106.04. Provider asked if she had questions or concerns regarding the method she currently uses, if she uses any methods<br><br>Q106.05. Client told about concerns about method, or asked questions about method, including possible side effects of method<br>Q106.07. Provider asked client if she has any questions<br><br>Q106.08. Provider and client talked about switching if she wants to stop using a method | *Female Survey*<br>Q318. When you obtained your [implant] were you told by the provider about side effects or problems you might have with a method to delay or avoid pregnancy?<br>Q319. Were you told what to do if you experienced side effects or problems?<br><br>Q320. At that time, were you told by the family planning provider about methods of family planning other than the [implant] that you could use?<br>Q321. At that time, were you told that you could switch to a different method in the future? | |

*(Continued)*

| Condition | DHS/ SPA | PMA | SARA |
|---|---|---|---|
| 6. Woman knows where and when to go for removal | Not captured | *Female Survey*<br><br>QIMP_302. At the visit when the implant was inserted, were you told for how long the implant would protect you from pregnancy?<br><br>QIMP_303. How long were you told?<br><br>QIMP_304. Were you told where you could go to have the implant removed? | Not captured |
| 7. Service is affordable or free | *Client Exit Interview*<br><br>Q304. Was the cost for services or treatments at this facility a problem?<br><br>IF YES, PROBE: Would you say this was a major problem or a minor problem? | *Client Exit Interview*<br><br>Q214. Did you pay any money for any of the family planning services you received or were provided today?<br><br><br>*Service Delivery Point Survey*<br><br>Q404. Do family planning clients need to pay any fees in order to be seen by a provider in this facility even if they do not obtain a method of contraception?<br><br>These may be consultation or registration fees charged to everyone who is seen in this facility or may be specific to family planning clients.<br><br>*Female Survey*<br><br>QIMP_307. Why were you not able to have your implant removed? | Not captured |
| 8. Implant removal data collected and monitored | Not captured | Not captured | Not captured |

*Available SARA indicators from the SARA Core Instrument.

wait time, hours/days of service. Condition 6—woman knows where and when to go for removal—was considered fully captured by one survey item in PMA's female questionnaire, which asked current implant users if they were told where they could go to have their implant removed, how long the implant protects against pregnancy, and duration of protection.

## Facility-based conditions

Of the six facility-based conditions of client-centered implant removal, three were categorized as completely captured, two as partially captured, and one as not captured through any one of the nine survey instruments. Specifically, conditions categorized as captured completely included supplies in-place (condition 2), systems in-place for managing difficult removals (condition 3), and counseling and reinsertion/switching offered (condition 5), while those captured partially included availability of competent and confident providers (condition 1) and service is affordable or free (condition 7). One condition was not captured across any instrument: implant removal data collected and monitored (condition 8).

All three facility-based surveys collected extensive information about the necessary supplies and equipment for implant removal services (condition 2), even if such questions were asked about the facility's inventory for family planning services more broadly—not specific to removal services. In addition to direct questions about supplies and equipment for implant insertion or removal, the SDP survey implemented by PMA also included direct questions about case-based availability of removal services on the day of the interview, which required supplies and equipment to be in-place for the service to be

reported as available (e.g., "If a woman came today needing her implant removed, could that service be provided to her today onsite?") Only one data source, the SDP survey implemented by PMA, measured the existence of operational systems for managing difficult removals, completely capturing this condition (condition 3; "Would someone at this facility know where to send her to have the implant removed?").

Finally, both the SPA and PMA survey instruments included questions that captured the extent to which reassurance, counseling, and reinsertion or switching were offered (condition 5), though facility-level measurement was only included in the SPA's observation of family planning consultation tool. The most frequently used measure to assess counseling content for management of one's contraceptive methods was use of the Method Information Index Plus (MII+), which asks about counseling on other methods, side effects, management of side effects, and the option to switch one's method [20]. These questions are asked to all women who receive family planning services, and are not specific to removal care. Measurement of counseling for method management was also ascertained at the individual-level, using the MII+ in SPA's CEI instrument, DHS's woman's survey, and PMA's female questionnaire, and via the Quality of Contraceptive Counseling short scale (QCC-10) [21] in PMA's CEI instrument. As with the facility-level measurement of counseling practices via SPA's observational tool, questions about the content of contraceptive counseling within individual-level surveys were asked among all family planning clients, not just those receiving removal services for their contraceptive implant.

Facility-based measurement was more limited for availability of competent and confident providers (condition 1), which was proxied through questions about (i) availability of removal services, including for deeply placed removals via the SDP survey of PMA, (ii) provider training in general family planning via the SARA instrument, and (iii) provider training and timing of training for implant insertion/removals via the SPA health worker instrument. While these questions asked about the existence of trained providers to proxy provider competence, no questions directly asked about provider confidence or willingness to remove implants, an important aspect of this condition. Similarly, measurement of service affordability or no-cost removals (condition 7) was captured partially through the survey instruments. Specifically, the SDP survey of PMA asked about fees charged to family planning clients, even if they are not obtaining a method of contraception. While this does not directly ask about costs for removal services, facilities that report charging family planning clients for services beyond method procurement may also charge clients for removal services, offering insight into the status of this condition for client-centered removal. In addition to indirect questions at the facility-level, the CEI of the PMA survey includes questions about if clients paid any money for the contraceptive services they were provided on the day of interview, which could be analyzed among those reporting method removal on the day of interview. PMA's female questionnaire directly asks current implant users who reported being unable to remove their method, despite attempting to do so, the reason for not having their method removed, with cost being one of the possible response options. Similarly, the CEI of the SPA survey asks clients directly if cost was a problem for their family planning visit.

Finally, regular collection of data and monitoring on implant removal (condition 8) was not captured in any of the nine survey instruments. Specifically, no survey instrument directly measured the number of removals requested or administered at the facility level.

## Discussion

Contraceptive implant removal services are critical to ensuring reproductive autonomy, particularly in sub-Saharan Africa where implants have quickly become the most or second-most commonly used method [3]. In this study, we summarized the availability of data sources across eight essential conditions for client-centered implant removal services, utilizing both population-based and facility-based survey tools widely used in sub-Saharan Africa. Our analysis was anchored in a conceptual framework aimed at enhancing access to high-quality removal services, recognizing the myriad reasons individuals are not able to stop using implants when they desire, even in contexts where these methods are widely available and used [6]. Of the eight client-centered conditions for implant removal proposed by Christofield and Lacoste, four were either only partially captured or completely absent from all nine survey instruments across three large-scale data collection

platforms. Results highlight critical measurement gaps that hinder researchers and programs from effectively monitoring and evaluating implant removal services, which are essential for safeguarding human rights in reproductive health.

Several conditions were fully captured in existing data sources, such as available supplies, systems for difficult removals, support for implant users, and awareness of removal timing and location. The availability of these data in existing survey tools offers promise for future research on health system environments that facilitate client-centered removal services. Recent research has identified gaps in facility readiness to provide removal services with these data, shedding light on potential health systems barriers to protecting users' reproductive autonomy to discontinue their methods [8]. While data on these four conditions is available, improvements are needed to enhance their utility for actionable program changes. For example, survey items related to reassurance and counseling primarily rely on the MII+ indicator, which may not adequately reflect the follow-up care necessary to support implant users and users of other provider-dependent methods. Similarly, assessment of whether systems are in-place for managing difficult removals (condition 3) provides information about whether any referral system exists, without further insights into whether they are functional or how they are used.

The singular condition that was not captured in any large-scale survey was implant removal statistics, such as the number of implants removed or number women referred for removal. Without such data, the reproductive health field is limited in its understanding of how often people seek implant removal, how frequently this service is provided, and how removals are differentially provided across diverse facility types—a data need highlighted by practitioners and researchers nearly a decade ago [6]. Timely removal data is essential to supporting national and subnational healthcare systems to monitor, project, and respond to population-level changes in demand for this service. These data should be included in health management information systems (HMIS)/DHIS2 and integrated into family planning registers and record books for regular data collection within facilities, as is done for implant insertions. There has been a growing effort to improve the quality and utility of health service statistics and shift away from reliance on large-scale surveys, highlighting an opportunity for this important addition [22,23]. A recent pilot study in Mozambique, published in 2022, demonstrated feasibility of integrating LARC removal indicators into HMIS and improved quality of care for LARC services [24]. Similar evaluation and scale-up efforts can help pave the way for ongoing monitoring efforts to strengthen LARC removal services.

The systematic lack of data on implant removal in existing population-based and facility-based surveys contrasts starkly with the available data on implant insertions (e.g., number of insertions provided, specificity about the type of implant, methods procured), which are typically ascertained from family planning registers as part of facility-based surveys. This imbalance in available metrics for monitoring implant removal versus method initiation highlights a gap in focus across the continuum of care for contraception. New, quality-centered measures of implant removal, including statistics that track the number of removals over time, can enable researchers and programs to have a more balanced picture of implant use dynamics and discontinuation. As the field strives to improve access to rights-based family planning services that ensure reproductive autonomy of contraceptive users and non-users, enhanced evaluation of implant removal is needed.

Our analysis also reveals a lack of comprehensive data on provider competence and confidence in providing implant removal services. Survey questions about provider training in implant removal can be used as a proxy measure for competence, but no included survey instruments queried provider's self-perceived competence, confidence, or willingness to remove implants. Capturing these provider-focused conditions are particularly important given that a growing body of work cites provider refusal to remove implants and intrauterine devices as one of the most commonly reported barriers to their timely discontinuation [7–15]. To enhance understanding of this condition, it is essential to develop new measures that assess provider confidence and willingness to facilitate implant removals. This is a critical step to supporting healthcare providers to deliver high-quality removal, while ensuring users can stop using their method when they desire.

Similarly, while population-based data provide some insight into individuals' knowledge of where they can go to seek removal, the single-item measure available through PMA surveys is limited. Lack of nuanced information on implant users' knowledge of when and where to go for removal limit the field's understanding of how equipped implant users are to exercise their reproductive choices, including decisions about when to discontinue a method. The inclusion of direct questions

about implant users' knowledge related to timing and location for removal in population-based surveys or client exit interviews could help fill this data gap and enhance understanding of resource needs for implant users.

Results should be interpreted in light of several limitations. First, while this mapping exercise sought to identify existing measurement of client-centered implant removal conditions in large-scale surveys, two of the surveys—SARA and SPA questionnaires—are not focused exclusively on reproductive health. The lack of available data on several conditions likely reflects the balance that many large-scale survey platforms make in terms of coverage across a wide array of health topics and depth into areas with high need for data. Second, many client-centered removal conditions, even those that were partially or completely captured by available survey items, may have still been captured insufficiently to measure the condition as comprehensively as necessary for programmatic intervention. For example, measurement of service affordability (one dimension of condition 7) in the existing surveys failed to capture informal fees that implant users might face when seeking removal, thereby limiting understanding of this condition. Finally, while this analysis—a text-based mapping exercise of survey items-to-indicators—provides critical insights into the coverage of client-centered implant removal indicators in existing large-scale data sources, it does not provide estimates for these indicators in any countries where survey data are available. While outside the scope of the present study, such an analysis would be a valuable contribution to the field to understand the current status of these client-centered removal conditions across contexts.

Despite these limitations, this mapping exercise serves as an essential step toward understanding the measurement landscape for implant removals. Using the most recent survey instruments available from three large-scale population-based and facility-based data sources, this research fills a critical gap in understanding how client-centered implant removal conditions are measured and where progress can be made to improve available data. To support reproductive autonomy of contraceptive users, it is essential that large-scale survey platforms and related reproductive health data sources integrate enhanced measures for monitoring client-centered implant removal services.

## Conclusion

Available, accessible, and high-quality implant removal services are central to protecting reproductive autonomy, including by supporting peoples' ability to decide when they want to stop using contraception. Following an established conceptual framework by Christofield and Lacoste, we find that measurement of client-centered implant removal services, knowledge, and access in existing population- and facility-based data sources widely used throughout sub-Saharan Africa is sparse, despite rapid growth in use of implants in the region over the past decade. Limited data available on the extent to which implant removal services are offered and provided (condition 8), whether removal services are affordable, free, timely and proximate for clients (condition 4 and condition 7), and whether providers are competent and confident in their ability to remove contraceptive implants (condition 1) warrants attention and further investigation to improve measurement. Nearly a decade after Christofield and Lacoste proposed these client-centered conditions, development and integration of standardized implant removal measures is urgently needed as an essential step toward protecting the reproductive autonomy of individuals who rely on these methods.

## Author contributions

**Conceptualization:** Celia Karp, Katherine Tumlinson, Leigh Senderowicz.

**Data curation:** Celia Karp.

**Formal analysis:** Celia Karp.

**Writing – original draft:** Celia Karp, Brooke W Bullington, Leigh Senderowicz.

**Writing – review & editing:** Celia Karp, Katherine Tumlinson, Brooke W Bullington, Linnea A Zimmerman, Leigh Senderowicz.

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
