## [Decision Letter · Decision Letter 0]

PGPH-D-25-00234

“Keeping the “R” in LARC (long-acting reversible contraception): Measuring quality-centered implant removal services in sub-Saharan Africa

Dear Dr. Karp,

Thank you for submitting your manuscript to PLOS Global Public Health. After careful consideration, we feel that it has merit but does not fully meet PLOS Global Public Health’s publication criteria as it currently stands. Therefore, we invite you to submit a revised version of the manuscript that addresses the points raised during the review process.

We will be requesting for an additional reviewer to ensure the improvement in the quality of your manuscript however, two reviewers have raised some comments which should be addressed. Address all comments in the attached files. 

We look forward to receiving your revised manuscript.

Kind regards,

Ifunanya Clara Agu

Academic Editor

Journal Requirements:

Additional Editor Comments (if provided):

Reviewers' comments:

Reviewer's Responses to Questions

**Comments to the Author**

1. Does this manuscript meet PLOS Global Public Health’s publication criteria?

Reviewer #1: Partly

Reviewer #2: Yes

2. Has the statistical analysis been performed appropriately and rigorously?

Reviewer #1: No

Reviewer #2: N/A

3. Have the authors made all data underlying the findings in their manuscript fully available (please refer to the Data Availability Statement at the start of the manuscript PDF file)?

Reviewer #1: Yes

Reviewer #2: Yes

4. Is the manuscript presented in an intelligible fashion and written in standard English?

Reviewer #1: Yes

Reviewer #2: Yes

Reviewer #1: The manuscript is technically sound and, in part, supported by the data. It describes methodologically rigorous research and is intelligibly presented. However, we would suggest the following improvements: I would suggest improvements to the manuscript, including:

1. Indicate whether a protocol has been submitted to an ethics committee and, if not, justify this omission. Also specify how ethical principles have been respected (informed consent, confidentiality of data, etc.).

2. Specify the analysis method used (descriptive statistics, thematic analysis, software used, etc.) to reinforce the scientific rigor of the manuscript.

3. Add concrete examples, quotations or figures to support the conclusions put forward.

4. Use Vancouver style for references to ensure a coherent presentation that conforms to scientific standards.

Reviewer #2: The manuscript fills an important gap in literature by evaluating how effectively current measurement tools of quality-centered implant removal services reflect Christofield and Lacoste's 8 conditions. It is well written and easy to read and follow. Sections follow the logical flow of scientific writing. There are very few edits I recommend making.

**Do you want your identity to be public for this peer review?** For information about this choice, including consent withdrawal, please see our Privacy Policy

Reviewer #1: No

Reviewer #2: No

---

## [Decision Letter · Decision Letter 1]

“Keeping the “R” in LARC (long-acting reversible contraception): Measuring client-centered implant removal services in sub-Saharan Africa

PGPH-D-25-00234R1

Dear Dr. Karp,

We are pleased to inform you that your manuscript '“Keeping the “R” in LARC (long-acting reversible contraception): Measuring client-centered implant removal services in sub-Saharan Africa' has been provisionally accepted for publication in PLOS Global Public Health.

Best regards,

Parvati Singh, PhD

Academic Editor

Reviewer Comments (if any, and for reference):

Reviewer's Responses to Questions

**Comments to the Author**

Reviewer #2: All comments have been addressed

Reviewer #3: All comments have been addressed

publication criteria?

Reviewer #2: Yes

Reviewer #3: Yes

3. Has the statistical analysis been performed appropriately and rigorously?

Reviewer #2: N/A

Reviewer #3: Yes

4. Have the authors made all data underlying the findings in their manuscript fully available (please refer to the Data Availability Statement at the start of the manuscript PDF file)?

Reviewer #2: Yes

Reviewer #3: Yes

5. Is the manuscript presented in an intelligible fashion and written in standard English?

Reviewer #2: Yes

Reviewer #3: Yes

Reviewer #2: All the comments I had made have been addressed by the author(s).

Reviewer #3: (No Response)

**Do you want your identity to be public for this peer review?** For information about this choice, including consent withdrawal, please see our Privacy Policy

Reviewer #2: No

Reviewer #3: No
